# Assessing the influence of climate on wintertime SARS-CoV-2 outbreaks

Rachel E. Baker [1,2 ✉], Wenchang Yang [3], Gabriel A. Vecchi [1,3], C. Jessica E. Metcalf [2,4] & Bryan T. Grenfell [2,4,5]

High susceptibility has limited the role of climate in the SARS-CoV-2 pandemic to date. However, understanding a possible future effect of climate, as susceptibility declines and the northern-hemisphere winter approaches, is an important open question. Here we use an epidemiological model, constrained by observations, to assess the sensitivity of future SARS-CoV-2 disease trajectories to local climate conditions. We find this sensitivity depends on both the susceptibility of the population and the efficacy of non-pharmaceutical interventions (NPIs) in reducing transmission. Assuming high susceptibility, more stringent NPIs may be required to minimize outbreak risk in the winter months. Our results suggest that the strength of NPIs remain the greatest determinant of future pre-vaccination outbreak size. While we find a small role for meteorological forecasts in projecting outbreak severity, reducing uncertainty in epidemiological parameters will likely have a more substantial impact on generating accurate predictions.

---

[1] High Meadows Environmental Institute, Princeton University, Princeton, NJ, USA. [2] Department of Ecology and Evolutionary Biology, Princeton University, Princeton, NJ, USA. [3] Department of Geosciences, Princeton University, Princeton, NJ, USA. [4] School of Public and International Affairs, Princeton University, Princeton, NJ, USA. [5] Division of International Epidemiology and Population Studies, Fogarty International Center, National Institutes of Health, Bethesda, MD, USA. ✉email: racheleb@princeton.edu

 **1**

The SARS-CoV-2 virus has spread across all geographic regions irrespective of local climate. Cases have continued to climb in both hot, humid conditions, such as the southern United States in summer and India during the southwest monsoon, and cold, dry conditions such as Wuhan province in China in the winter. High susceptibility has likely limited the role of climate in the early pandemic such that the signature of seasonality is not yet visible[1]. However, as susceptibility starts to decline, particularly in regions with high numbers of cases, the extent to which the climate may determine the future pandemic trajectory remains unclear.

Many directly transmitted infectious diseases display seasonal cycles of incidence[2]. For several viral infections, including varicella, influenza, and respiratory syncytial virus (RSV), these cycles have been shown to be dependent on climate[3–7] (along with seasonal population aggregation such as school terms). Specific humidity, in particular, has been shown to be important for influenza and RSV transmission[3,4,7], with low specific humidity correlated with increased virus survival for influenza. Evidence suggests that humidity may also play a role in determining airborne droplet size and hence residence time in the air[8]. Endemic coronaviruses have also demonstrated sensitivity to the climate in both laboratory studies[9] and at the population level[1,10]. While the novel coronavirus, SARS-CoV-2, also appears to be sensitive to the climate in laboratory settings[11,12], case data have yet to reveal a clear environmentally driven trend[13].

In a recent study, we used an epidemiological model to explore the subtleties of pandemic seasonality[1]. We showed that the climate plays a secondary role compared to high susceptibility in determining early pandemic trajectories, yet we identified potential climate impacts as susceptibility waned. Thus, a more detailed investigation of possible wintertime effects is important, especially given the major, locally variable impacts of non-pharmaceutical interventions (NPIs). Here we consider how climate conditions in the coming months, and pre- any major roll out of vaccines, may influence the future trajectory of the pandemic as susceptibility declines.

We probe the possible effect of climate while varying two factors: the level of depletion of susceptibles and the relative efficacy of NPIs in reducing transmission. Building on our prior work[1], we use a climate-driven Susceptible-Infected-Recovered-Susceptible (SIRS) model to simulate the disease dynamics under these different scenarios and across different climates. The model is based on the estimated climate sensitivity of endemic betacoronavirus HKU1 (results for betacoronavirus OC43 are shown in Supplementary Figs. 6 and 7). This betacoronavirus was found to be more sensitive to climate in our recent work and so our simulations reveal the upper bound on a possible climate effect.

A key question is the extent to which the upcoming northern hemisphere winter climate may exacerbate future cases numbers. To address this, we first consider possible case trajectories for New York City (results for select other northern hemisphere locations are shown in Supplementary Figs. 1–3).

## Results

**Wintertime outbreaks in the northern hemisphere.** In Fig. 1a we use case data (see "Methods") to estimate the effective reproductive number of infection for New York City from the start of 2020 to the present (July 2020)[14]. Estimated values of $R_{effective}$ peak early in the outbreak and then settle close to 1 in the summer months as NPIs act to lower transmission. We assume the $R_{effective}$ values approximate $R_0$ and compare them to the predicted seasonal $R_0$, derived from our climate-driven SIRS model. The model assumes the climate sensitivity of betacoronavirus HKU1 and that seasonal variations in transmission are

driven by specific humidity. Current rates (average over second and third weeks of July) of $R_{effective}$ in New York city are found to be approximately 35% below the $R_0$ levels predicted by our climate-driven model. We assume this 35% decline is due to the efficacy of NPIs. To project future scenarios we assume that $R_0$ remains at either the current levels (constant) or a relative 35% decrease in our climate-driven $R_0$, which means $R_0$ oscillates with specific humidity (Fig. 1a, top plot).

In Fig. 1a (lower plots) we show the proportion infected over time using the climate-driven and constant $R_0$ values. We also vary the reporting rate of observed cases relative to modeled cases; while this accounts for under-reporting it also allows us to vary the proportion susceptible over a feasible range (see "Methods"). In the middle figure, the reporting rate is 10% (estimates for US reporting rates are <10%[15]), which implies a relatively small reduction in susceptibility based on case data pre-July. In this case, a small boost to transmission, driven by low specific humidity in the winter months, results in a relatively large secondary outbreak in the climate scenario. In the constant scenario, $R_0$ stays below 1 and there is no outbreak in the winter months. We also consider a scenario where the reporting rate is 3% (Fig. 1a, lower plot). In this case the lower reporting rate means more cases (relative to the observed case counts) and a greater reduction in susceptibility. This results in a smaller wintertime outbreak in the climate scenario.

In Fig. 1b we consider a scenario where NPI measures are relaxed further such that $R_0$ is reduced 15% below non-control values as of the last week in July. In this case $R_0 > 1$ for both the climate and constant scenario and case numbers begin to grow exponentially. With a 10% reporting rate a large secondary outbreak is observed in both the constant and climate scenarios (Fig. 1b, middle plot). With a 3% reporting rate, meaning a larger depletion of susceptibles, the secondary outbreak appears much larger in the climate scenario: this supports the hypothesis that the disease will become more sensitive to climate as the susceptible proportion declines, much like the seasonal endemic diseases.

In Fig. 1c–h we simulate model outcomes across a broad range of parameter space varying the proportion susceptible (in July) and the reduction in $R_0$ due to NPIs. The proportion susceptible is varied by initializing the epidemic with different sizes of the infected population (initializing with a large number results in a relatively larger outbreak and initializing with a small number results in a smaller outbreak). We vary this starting number over a feasible range given the case data, i.e., such that observed cases never exceed modeled cases or that the reporting rate never drops below 1%. Over this range, the model plausibly tracks the observed case data.

Figure 1e shows the change in winter peak size (max proportion infected between September–March) due to climate. Peak size results for the constant and climate scenarios are shown in Fig. 1c and d, respectively. When the susceptible proportion is high and the effect of NPIs are minimal (relative $R_0$ given NPI = 1), large outbreaks are possible in both the climate and constant $R_0$ scenarios meaning the relative effect of climate on peak size and timing is close to 0 (top right Fig. 1e). As the proportion susceptible declines (moving left along the x-axis of Fig. 1e), case trajectories become more sensitive to the wintertime weather resulting in larger peaks in the climate scenario. However, sufficiently strong NPIs, in combination with low susceptibility, reduce incidence to zero in both the climate and control scenarios (bottom left Fig. 1e). NPIs are not as effective at reducing cases when susceptibility is higher (bottom right Fig. 1e).

We also consider the effect of climate on secondary peak timing. Figure 1f, g shows the peak timing in years (relative to

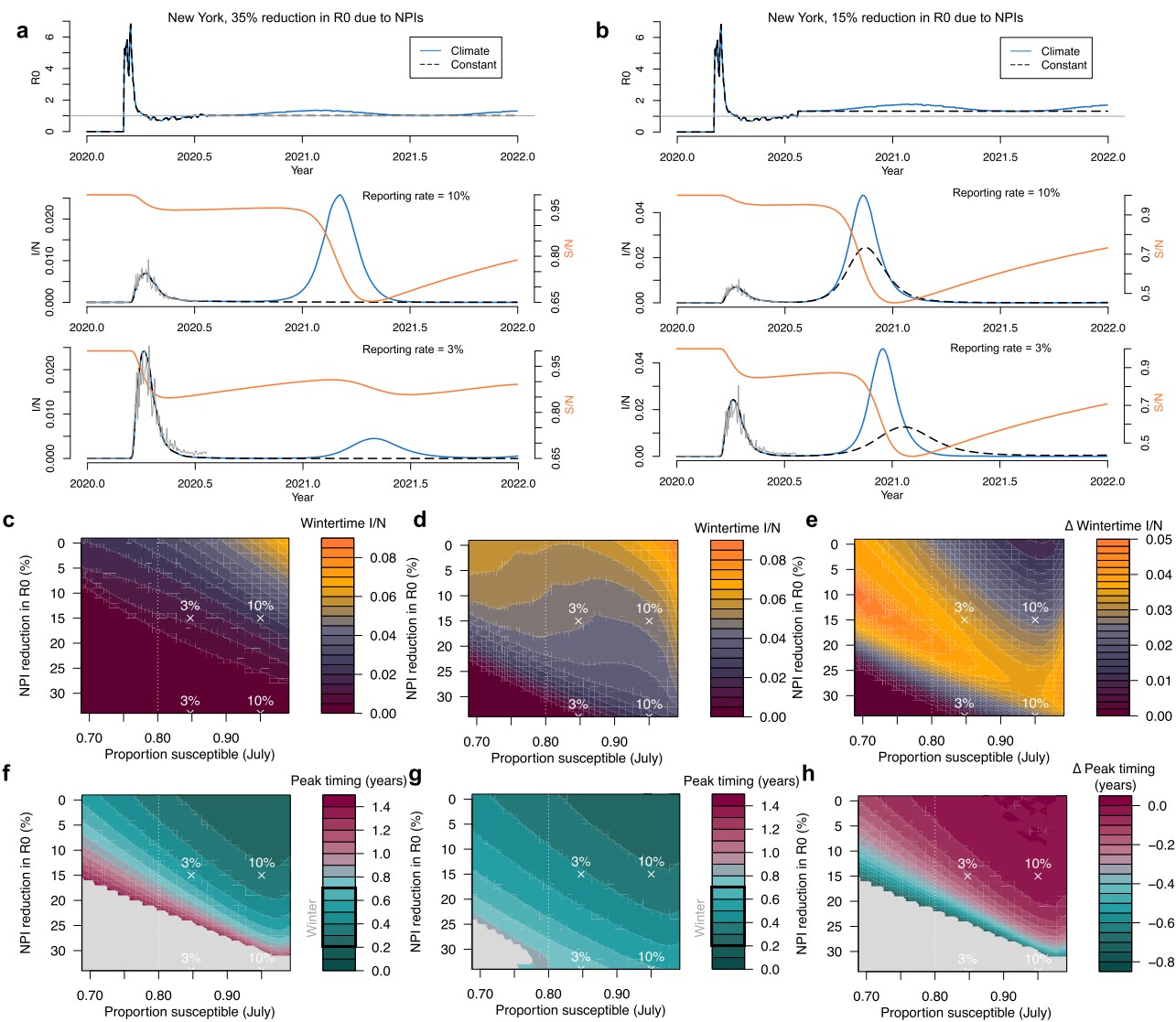

**Fig. 1 Wintertime outbreaks in New York City.** Estimated and projected $R_O$ values (top plot) assuming **a** 35% and **b** 15% reduction in $R_O$ due to NPIs. Corresponding time series show the simulated outbreaks in the climate (blue) or constant (black/dashed) scenarios, with middle row plots assuming a 10% reporting rate and bottom row plots assuming a 3% reporting rate. Corresponding susceptible time series are shown in orange (susceptibles = $S$/population = $N$). Case data from New York City are shown in gray. Surface plots (top) show the peak wintertime proportion infected (infected = $I$/population = $N$) in the scenarios with **c** the constant $R_O$ and **d** the climate-driven $R_O$. **e** shows the difference between the climate and constant $R_O$ scenario. The timing of peak incidence in years from July is shown for the **f** constant and **g** climate scenarios. The difference between climate and constant scenario is shown in **h**. Points in **c**–**h** show the scenarios is **a**, **b**. Dashed line shows estimated susceptibility in New York based on ref. [24].

July 2020) in the constant and climate scenarios, respectively. In the climate scenario, peak timing for New York is clustered in the winter months (Fig. 1a, b). In the constant $R_O$ scenario, secondary peaks can occur at a wide range of times over the next 1.5 years. As in the peak size results, high susceptibility and limited NPIs reduce the effect of climate and peak timing is matched for both the climate and control scenarios (top right Fig. 1h). Gray areas represent regions where there is no secondary peak in either the climate or control scenario.

**Climate effects on global risk.** We next consider the relative effect of climate on peak size for nine global locations (Fig. 2b). In this case, as opposed to using estimated $R_{effective}$ values (given case data are not available for several of the global cities), we simulate the epidemic from July 2020 using a fixed number of infecteds and vary the starting proportion of susceptibles

(example results from select global locations, using estimated $R_{effective}$, are shown in Supplementary Figs. 1–3). Results from the New York surface in Fig. 2b qualitatively match our tailored simulation in Fig. 1. Locations in the southern hemisphere are expected to be close to their maximum wintertime $R_O$ values in mid-2020 (Fig. 2a), meaning that secondary peaks in the climate scenario are lower than the constant $R_O$ scenario for these locations (Fig. 2b). Tropical locations experience minimal difference in the climate versus constant $R_O$ scenario given the relatively mild seasonal variations in specific humidity in the tropics. Broadly, the results across hemisphere track the earlier results from New York: high susceptibility and a lack of NPIs lead to a limited role of climate, but an increase in NPI efficacy or a reduction in susceptibility may increase climate effects. This result is more striking in regions with a large seasonality in specific humidity (e.g. New York, Delhi and Johannesburg).

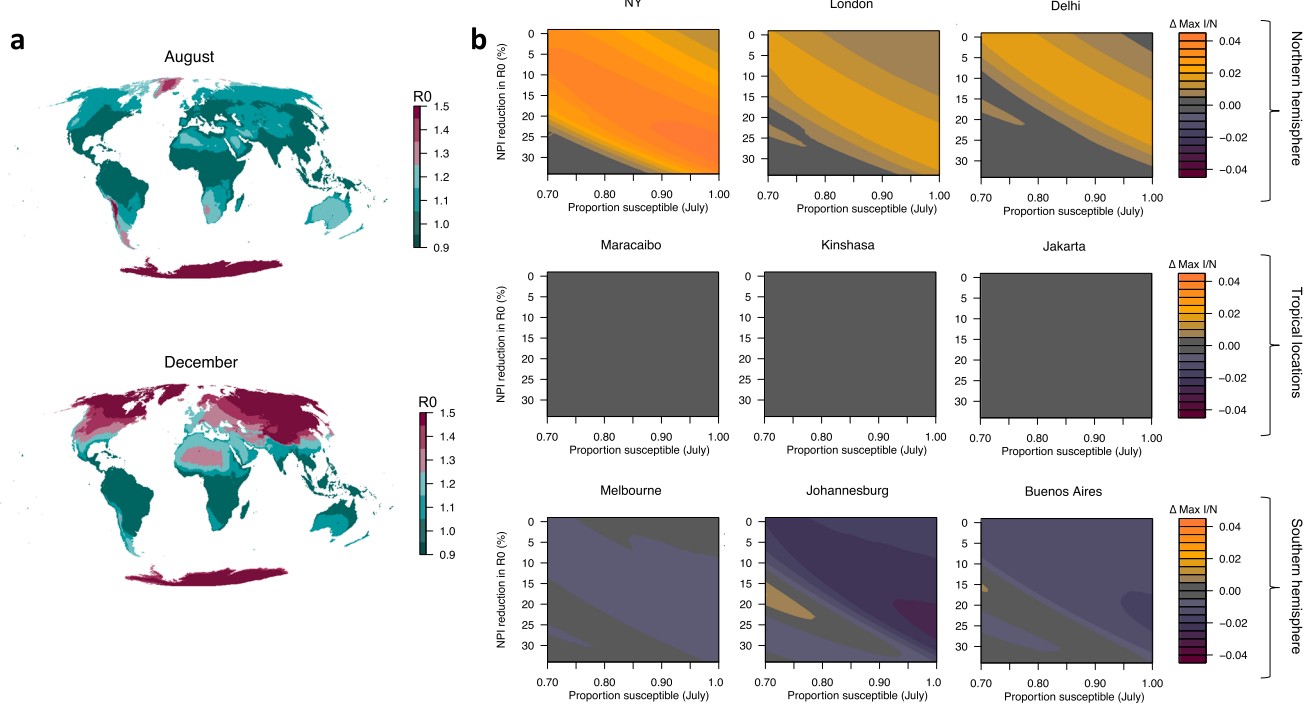

**Fig. 2 Climate sensitivity of outbreaks across global locations. a** The climate effect on $R_0$ assuming a 35% reduction due to NPIs shown for August and December. **b** The effect of climate, changing susceptibility, and NPIs on peak proportion infected (infected = $I$/population = $N$), post July 2020, for nine global locations.

**Drivers of variability in secondary outbreak size.** Our results suggest that climate may play an increasing role in determining the future course of the SARS-CoV-2 pandemic, depending on levels of susceptibility and NPIs. We next evaluate the extent to which interannual variability in specific humidity could influence peak size. We simulate separate New York pandemic trajectories using 11 years (2008–2018) of specific humidity data. Figure 3a shows the variability in $R_0$ and secondary peak size based on these runs (with 35% reduction in $R_0$ due to NPIs and 10% reporting rate—the same as Fig. 1a). While a relatively large peak occurs in all years, the largest peak (0.038 proportion infected) is almost double the smallest peak year (0.020 proportion infected). In Fig. 3b we calculate the coefficient of variation of the peak size for different susceptible proportions and NPI intensities. These results qualitatively track Fig. 1e. Sensitivity to interannual variation appears most important when the susceptible population has been reduced by at least 20% and minimal controls are in place.

Many factors, including weather variability, determine the size of a possible secondary outbreak. Another factor that may play an important role is the length of immunity to the disease. While the length of immunity may not affect the dynamics in the early stage of the pandemic, it could have complex and uncertain outcomes for future trajectories[16]. In our main results, we assume a length of immunity equal to betacoronvirus HKU1, based on prior estimates[1]. We also assume a climate sensitivity based on estimates for HKU1. However parameters for SARS-CoV-2, such as immunity length and climate sensitivity, are still fundamentally uncertain.

We consider the possible contribution of uncertainty in parameters to the variance in the wintertime peak size following the method developed by Yip et al.[17] (see "Methods"). We run our simulation for New York while varying parameter values for the efficacy of NPIs, the length of immunity to the disease, the reporting rate of prior cases (which defines susceptibility in July), the climate sensitivity of the pathogen (in terms of the strength of the relationship with specific humidity), and the weather variability (interannual variability determined by historic weather observations from a particular year, 2009–2018). We then perform an analysis of variance (ANOVA) on the determinants of wintertime peak size.

Figure 4 shows contribution to variance in wintertime peak size of these five parameters: NPIs efficacy, immunity length, reporting rate, climate sensitivity of the virus, and interannual weather variability. We find that climate sensitivity is an important factor but secondary to the efficacy of NPIs and immunity length in determining peak transmission. Uncertainty in immunity length and reporting together influence susceptibility and collectively account for the second largest portion of total uncertainty. Uncertainty in interannual variability, i.e. weather, has a smaller impact on peak size. NPIs contribute the largest proportion to total variance in peak size. It is important to note that while other parameters are external features of either the virus, climate, or disease trajectories to date, the efficacy of NPIs is determined directly by policy interventions and therefore the size of future outbreaks is largely under human control.

## Discussion

Our results suggest that NPIs remain the primary determinant of future SARS-CoV-2 outbreak size. However, in a highly susceptible population, with NPIs in place that keep $R_0$ just below 1, a small boost to transmission due to wintertime climate conditions could be sufficient to drive a large outbreak. In this case, more stringent NPIs may be required in winter months to limit such an outbreak. In all cases, if susceptibility is high, and NPI measures are reduced, large outbreaks will occur no matter the climate conditions.

There are several caveats to our results. First, the precise mechanism by which climate modulates seasonal transmission

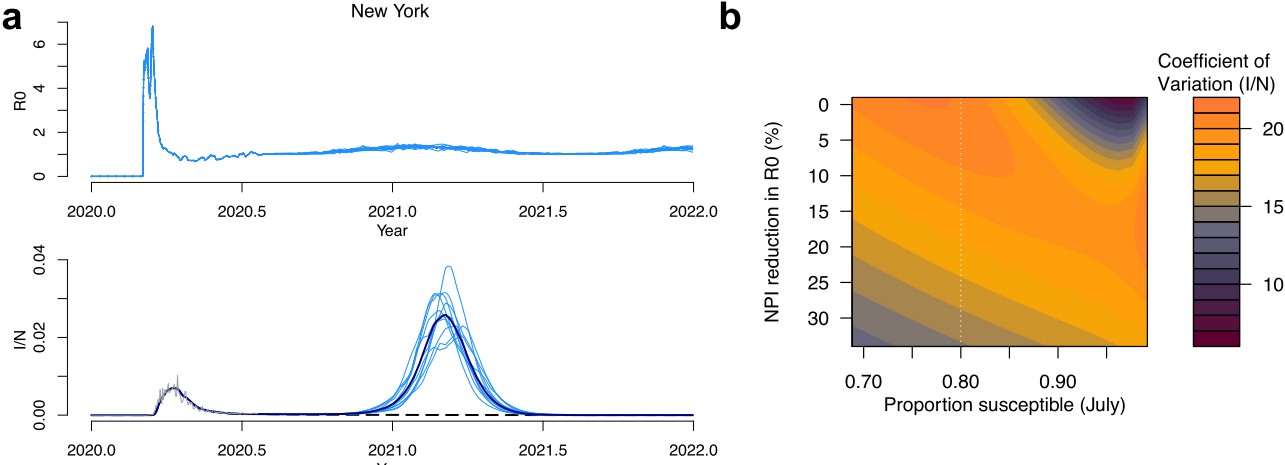

**Fig. 3 Climate variability and wintertime cases in New York. a** Climate-driven $R_O$ and corresponding infected time series (infected $= I$/population $= N$) based on the last 10 years of specific humidity data for New York, assuming a 35% reduction due to NPIs. **b** The effect of changing susceptibility and NPIs on the coefficient of variation of peak incidence for simulations using specific humidity data from 2008 to 2018. Dashed line shows estimated susceptibility in New York based on ref. [24].

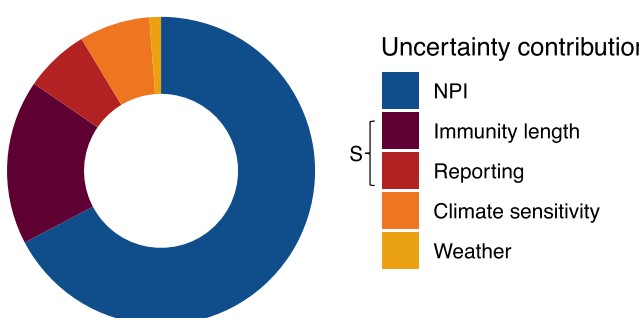

**Fig. 4 Contribution to uncertainty in New York wintertime 20/21 peak size.** The relative importance of NPI efficacy [0–35%], immunity length (10–60 weeks), reporting (1–100%), climate sensitivity of the virus [−32.5 to −227.5], and interannual weather variability [10 years] in determining wintertime peak size. Immunity length and reporting rate collectively determine susceptibility, $S$.

rates for viruses is currently unknown. While virus survival and changes to the immune system are expected to fluctuate with the weather, changes to human behavior, such as grouping indoors during cold weather, may partly determine the seasonal effect. Given the broad societal disruptions of the COVID-19 pandemic, these latter behaviors are likely to be reduced, such that total climate-driven fluctuations to transmission may be modified. Further, additional seasonal behaviors that may have also driven transmission, such as population aggregation through schooling, will also be reduced by NPIs.

Second, we do not directly estimate the climate sensitivity of SARS-CoV-2. Studies exploring this relationship using case data have yet to find a definitive result; however, there is growing consensus that the virus is moderately affected by cold, dry conditions[12,13]. Instead our model relies on estimates of the climate-sensitivity of another betacoronavirus, HKU1. HKU1 was the most sensitive out of the two betacoroanviruses explored in our earlier work[1], so our results likely present the upper bound of a possible climate influence. Simulations using the less climate-sensitive coronavirus, OC43, suggest a smaller effect of climate on wintertime peak size (Supplementary Fig. 6). In the OC43 scenario, $R_0$ remains high throughout the year with less of a decline in the summer months. This means the measured NPI efficacy is

greater in the OC43 scenario: NPIs reduce $R_0$ by 55% below what it would have been in the summer months to reach $R_0 \simeq 1$, as opposed to the 35% reduction in the HKU1 scenario. In Supplementary Fig. 7 we run simulations using OC43 parameters over this larger range of NPI efficacy (0–55%) and recover a qualitatively similar pattern to our HKU1 results, although with a reduced maximum climate effect ($\Delta I/N = 2\%$ as opposed to 5% for HKU1).

Third, evidence from other respiratory viral pathogens implies that tropical locations may experience distinct climate drivers and dynamics of infection[7,18,19]. Given our model is parameterized by fitting to US data, we may not have captured the full suite of possible climate drivers in these locations. While data from tropical climates are limited, evidence from Malaysia suggests persistent year-round infections for the endemic betacoronaviruses[20], a feature our model is able to capture (Supplementary Fig. 9). However, possible secondary climate drivers will not be accounted for the model and could bias our results for tropical locations. We also note the US betacoronavirus data are at a course spatial resolution with a short time horizon, limiting our ability to identify the signal of interannual specific humidity variability on cases. A high correlation in the seasonality of specific humidity and temperature (Supplementary Fig. 10) limits our ability to separately identify the effect of these two variables. Fitting our model to temperature data would like give similar results.

Our results imply that meteorological and climate forecasts could be helpful in predicting future outbreak size (Supplementary Fig. 5). However, this information will likely be secondary to epidemiological monitoring such as estimates of the efficacy of active control measures in reducing transmission and serological surveys to determine susceptibility[21]. Titrating the impact of ongoing and future vaccination programs on susceptibility, and hence climate, will be important in the coming months. Current data from serological surveys suggest a minimal reduction in susceptibility in many locations[22], with New York City towards the upper bound in terms of total reduction in susceptibility[23,24]. For many other locations, susceptibility may be much closer to 1, meaning the efficacy of NPIs will be a key determinant of winter outbreak size. While our model assumes a constant reduction in $R_0$ due to NPIs, in reality policy makers likely adapt to rising case numbers by enforcing stricter measures. For instance, we

estimated an $R_0 > 1$ in Victoria, Australia in July and projected a much larger outbreak than has been observed to date (Supplementary Fig. 4). A second lockdown, enacted in Australia in August, substantially curbed this outbreak. Our results therefore suggest that more stringent NPIs may be required during the winter months to minimize total risk.

## Methods

**Data.** Global COVID19 case data come from the Johns Hopkins coronavirus resource center (https://github.com/datasets/covid-19, country-level data)[25]. County-level coronavirus cases data, including estimates for New York city, come from the New York Times (https://raw.githubusercontent.com/nytimes/covid-19-data/master/us-counties.csv, US county-level data). Specific humidity data come from ERA5 (ref. [26]). The shapefile used for map outlines in Fig. 2a comes from thematicmapping.org and is available under Creative Commons Attribution-Share Alike License 3.0.

**$R_0$ estimates.** We use the EpiEstim package in the R programming language to estimate $R_{effective}$ from coronavirus cases data assuming an uncertain serial interval with a mean of 4.7 days and a standard deviation of 2.9 days. We calculate $R_{effective}$ estimates from the first date in 2020 where case numbers are greater than zero for a particular location. $R_{effective}$ is estimated until 21 July 2020 (the date we first accessed the data). After 21 July 2020 we use an $R_0$ value modulated by climate and NPI efficacy. The climate-driven $R_0$ values are based on the climate-driven SIRS model[1]. Specifically, climate-driven values of $R_0$ are given by

$$R_0(t) = \exp(a * q(t) + \log(R_{0\,max} - R_{0\,min})) + R_{0\,min} \qquad (1)$$

where $R_{0max}$ and $R_{0min}$ are the maximum and minimum reproductive numbers, respectively, set at 2.5 and 1.5 (refs. [1,27]). Most estimates of $R_0$ for SARS-CoV-2 lie in the range of 2–3 (refs. [28–30]), though some studies have estimated an $R_0$ as high as 6.49 and as low as 1.44 (ref. [28]). We set $R_{0max} = 2.5$ as a conservative upper bound. In prior work we set $R_{0min} = 1.5$ to reflect the 40% reduction in transmission observed due to climate in studies of other viruses[1,27]. We find this value matches lower bound estimates of $R_0$ (ref. [28]); however, the strength of the climate effect for SARS-CoV-2 remains uncertain. Results using a higher $R_{0max}$ are shown in Supplementary Fig. 8.

$q(t)$ is specific humidity and $a$ is the climate dependence parameter (set at $-227.5$) based on model fits for the HKU1 betacoronavirus[1]. We test sensitivity of our results to different values of parameter $a$ in Fig. 4 and Supplementary Figs. 6 and 7. Importantly, the short time series available on the endemic coronaviruses and course spatial resolution of the data limits our ability to identify the effect of interannual variation in specific humidity on interannual variation in cases. We fit our model to the mean seasonality in specific humidity which follows a sinusoidal pattern tightly correlated with temperature (Supplementary Fig. 10). Fitting to temperature would likely give similar results (though with scaled values of $a$). Other climate variables seasonally correlated with specific humidity, such as patterns of ultraviolet radiation, may have a similar effect. However, we retain specific humidity as a driving variable due to prior understanding of its role in respiratory pathogen transmission[3,4,7].

We define the NPI efficacy as a percentage reduction in $R_0$ below the levels predicted by Eq. (1). For example, in New York in July we estimate an average $R_0$ of 1.04. In comparison Eq. (1) predicts an $R_0$ of 1.61 for this time period. Therefore, we assume NPIs have $1 - 1.04/1.61 = 35\%$ efficacy. In the climate scenario we project $R_0$ forward at a 35% reduction of the calculated value of Eq. (1). In the constant scenario, we assume $R_0$ remains constant at 1.04. We repeat this exercise while varying percentage reduction, i.e., NPI efficacy, in Fig. 1c–h.

**SIRS model.** Our $R_0$ estimates are incorporated into an SIRS model where $R_0(t) = \beta(t)D$. $D$ is the mean infectious period (set at 5 days) and $\beta_{(t)}$ is the contact rate. The SIRS model is directly dependent on $\beta_{(t)}$ and is given by

$$\frac{dS}{dT} = \frac{N - S - I}{L} - \frac{\beta(t)IS}{N} \qquad (2)$$

$$\frac{dI}{dT} = \frac{\beta(t)IS}{N} - \frac{I}{D} \qquad (3)$$

where $S$ is the number of susceptibles, $I$ is the number of infecteds, and $N$ is the population size. $N = S + I + R$, where $R$ is the number of individuals in the recovered category.

We initialize the model on the first day cases are observed. In order to capture different possible trajectories, we initialize varying the proportion infected on the first day. Over a finite range, models initialized with different infected proportions are able to track observed cases to a scaling constant, i.e., the reporting rate (Fig. 1a, b). We tune the range of starting proportion infected such that the reporting rate stays between 1 and 100%, though results for specific trajectories are shown for 3 and 10% in Fig. 1a, b, reflecting prior estimates on reporting rates[15].

**Uncertainty decomposition.** We run our model for New York using ten discrete values of non-pharmaceutical intervention, immunity length, reporting rate (determined by the initial proportion infected), climate sensitivity, and weather variability (determined by using historic weather observations from 2009 to 2018): a total of 10,000 model runs. For each model run, we record the wintertime peak size. We then use ANOVA on a fixed effect regression model where the dependent variable is wintertime peak size and the fixed effects are the factorial contribution of each parameter. The total sum of squares is calculated for each parameter across factors. A similar approach has been used to decompose uncertainty in climate model projections[17]. Using fixed effects allows us to recover some of the non-linearity in possible parameter dependence.

To create Fig. 4, we divide the sum of squares attributable to each parameter by the total explained sum of squares. Parameters were varied over a plausible range, i.e., NPI efficacy [0–35%], immunity length [10–60 weeks], reporting (1–100%), climate sensitivity (OC43 climate sensitivity of $-32.5$ to HKU1 climate sensitivity of $-227.5$ (ref. [1])) and weather variability (based on 2009–2018 weather). However, it is important to note that expanding the range of a particular parameter would likely increase the importance of its predicted effect. As such, this method only provides a proxy for considering possible contributions to uncertainty.

**Reporting summary.** Further information on research design is available in the Nature Research Reporting Summary linked to this article.

## Data availability
Coronavirus case data, for estimating $R_{effective}$, were downloaded from the New York Times (https://raw.githubusercontent.com/nytimes/covid-19-data/master/us-counties.csv, US county-level data) and Johns Hopkins coronavirus resource center (https://github.com/datasets/covid-19, John Hopkins, country-level data)[25]. Specific humidity come from ERA5 [https://www.ecmwf.int/en/forecasts/datasets/reanalysis-datasets/era5][26].

## Code availability
Code for recreating the main results is available via github at https://zenodo.org/record/4323552 (ref. [31]).

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

## Acknowledgements
This study is supported by the Cooperative Institute for Modeling the Earth System (CIMES) and the High Meadows Environmental Institute.

## Author contributions
Conceptualization, methodology, and writing, reviewing, and editing: R.E.B., W.Y., G.A.V., C.J.E.M., and B.T.G.; data curation: R.E.B. and W.Y.; and formal analysis, software, visualization, and writing, original draft: R.E.B.

## Competing interests
The authors declare no competing interests.
