## [Peer Review File · Nature Communications]

REVIEWER COMMENTS

Reviewer #1 (Remarks to the Author):

Convincing study of the potential effects of absolute humidity on Covid-19 transmission dynamics, allowing for the effects of NPI and population susceptibility.

The meaning of Figure 2 could be made clearer by editing the caption as follows

The effect of *climate*,
changing susceptibility and NPIs on peak proportion infected...

and in line 123:

... meaning that secondary peaks in the climate scenario are lower than the constant R0 scenario for these locations *(add cross reference to Fig 2b)*

Further discussion of the implications of the model for Southern Hemisphere countries would be of interest.

Discussion

Consider commenting on the proposed physical explanations for effects of absolute humidity on viral transmission. In particular, airborne transmission of respiratory viruses is thought to be enhanced by desiccation of exhaled particles in dry air. This decreases particle aerodynamic diameter and thereby increases airborne residence time [3,4].

Minor comment

NPI= non pharmaceutical intervention, rather than control

Ref 10 is incomplete

Simon Hales

Reviewer #2 (Remarks to the Author):

Summary: The authors use an epidemiological model for SARS-CoV-2 with an assumed climate-dependent transmission rate to project the outbreak into the future in New York City and other locations under varying levels of non-pharmaceutical interventions and population susceptibility. This is a clearly written paper on a topic of great interest and importance, but I have some concerns with it in its current form.

Major comments: My main concern with the study as it is currently written is with regards to some of the underlying assumptions regarding climate sensitivity of SARS-CoV-2, and the lack of justification and exploration of the importance of these assumptions.

1) Assuming climate sensitivity of SARS-CoV-2 is the same as HKU1

The first and most important assumption from my perspective is the assumption that the climate sensitivity of SARS-CoV-2 will match that as estimated for another betacoronavirus, HKU1. The introduction (lines 52-53) notes to "see Discussion for the effect of varying climate sensitivity", but I could only find a sentence or two noting that there has yet to be a conclusive result from studies exploring the climate sensitivity of SARS-CoV-2 and so the authors have assumed it follows that of HKU1. In the authors' previous study that this work builds on (Baker et al 2020, Science), they

estimate that HKU1 shows the strongest sensitivity to climate of the three viruses compared, while the betacoronavirus OC43 shows the weakest climate sensitivity. If these two viruses represent the two extremes (strong climate sensitivity versus weak climate sensitivity), are there reasons that we should expect SARS-CoV-2 to match the pattern of strong climate sensitivity? If, instead, it's equally likely to follow the weak climate sensitivity of OC43 (or to fall somewhere in the middle), I think these scenarios should be given more exploration here. I appreciate that the authors have done the uncertainty decomposition and have included varying climate sensitivity as a factor in Figure 4, but I don't think this necessarily makes clear the importance of this assumption or shows the effect that assuming a different climate sensitivity would have on the results. In other words, I'm wondering how different Figure 1 and the results section would look if the authors had assumed SARS-CoV-2 follows the climate sensitivity of OC43 instead of HKU1?

2) R0max and R0min

I was also hoping for a bit more justification for the assumed R0max and R0min values used in Eq. 1 beyond the explanation that they were previously estimated in the Baker et al 2020 and Kissler et al 2020, as the choices for these values can greatly affect how strong the realized climate effect is on the virus. From my reading of Baker et al 2020, it seemed that R0max of 2.5 was used as a conservative lower estimate early in the epidemic; does this still hold as the likely maximum R0 value for SARS-CoV-2? Similarly, it seems that R0min of 1.5 was used because other viruses have shown up to a 40% decrease due to environmental factors. Do we have any reasons to suspect that SARS-CoV-2 follows this same 40% pattern? Overall, even though it may be somewhat repetitive of what was stated in the Baker et al 2020 Science paper, I think this section could benefit from some added methods descriptions that state the assumptions underlying this equation.

3) Application of the model to global locations and potential alternative drivers

It is not clear to me if the same model used to predict wintertime outbreaks in New York City should be applied to a range of global cities, including some in the tropics. From what I understand, the climate sensitivity relationship (Eq. 1) that is applied here to global locations (Fig. 2) was fit in Baker et al 2020 Science to surveillance data from the United States and is driven by changes in specific humidity. While specific humidity may be the main climate driver of betacoronavirus dynamics in the United States or temperate regions, is it a fair assumption that it is also the main climate driver in some of the tropical cities modeled here in Figure 2? For example, in Tamerius et al 2013 (PLoS Pathogens), they found that while influenza cases peaked under cold and dry conditions in temperate areas, tropical areas showed a different, u-shaped response where influenza cases peaked at very low levels of specific humidity but also at very high levels of specific humidity. It therefore seems very possible that other climate drivers that have been suggested to be important in the tropics for transmission of other respiratory viruses like influenza, such as precipitation or ultraviolet radiation, could be more important factors than specific humidity for SARS-CoV-2 climate dependence in some of the cities modeled here. The authors state that "tropical locations experience minimal difference in the climate versus constant R0 scenario given the relatively mild seasonal variations in specific humidity in the tropics", but can they please provide some justification for why they expect specific humidity to be the climate driver in these cities, and why they'd expect to see the same transmission - specific humidity relationship that was estimated in the United States hold across the globe?

Relatedly, given that other climate drivers may be relevant and have similar average seasonal patterns but different interannual variation, is there evidence that the interannual variation in specific humidity drives interannual variation in other coronaviruses? The authors argue that there could be large effects of interannual variation in specific humidity in NYC on interannual changes in cases, but the climate-sensitivity is estimated using the average seasonal pattern of specific humidity. Alternatively, if there's no signal of an effect of interannual variation of specific humidity on interannual variation in HKU1 or OC43 cases over the last years, should we expect an effect for SARS-CoV-2?

Minor comments

Line 20: if the abbreviation is NPIs, I think this should read "interventions" instead of "controls".

Line 22-23: "Our results imply a role for meteorological forecasts in projecting outbreak severity". Does this statement rely on the assumption that SARS-CoV-2 climate dependence will be as strong as HKU1? If SARS-CoV-2 climate dependence instead is similar to OC43, would meteorological forecasts still be useful?

Lines 37-38: "Endemic coronaviruses have also demonstrated sensitivity to the climate in both laboratory studies [8] and at the population level [7]." Reference 7 relates to RSV – are there references for coronavirus that could be used here?

Lines 44-45, 53-54: The emphasis on effects in "northern hemisphere" locations seems misplaced given the broader focus on global cities later in the paper.

Lines 61-64: Further explanation of the "climate-driven model" would help here to explain the assumptions inherent in the subsequent results.

Lines 109-110: Is this result that secondary peak timing for New York is clustered in the winter months in the climate scenario (but can occur at a wide range of times in the constant scenario) almost a foregone conclusion because of the way climate-dependence is incorporated into the authors' model?

Lines 127-128: "This result is more striking in regions with a large seasonality in specific humidity" – can the authors add a note to the end of this sentence (and/or in the Fig. 2 caption) which cities this refers to specifically.

Lines 130-140, 168-169: Is there any analogous evidence from the other betacoronaviruses to help support the argument that interannual variation could have large effects on interannual changes in cases? See major comment 3 for more on this.

Line 193: typo in "sensitivity".

Lines 198-200: L in Eq. 2 is not defined in text, and R, which is defined, does not appear in either equation. I assume it's just that $N = S + I + R$, but that would be worth stating.

Line 199: I think there's a typo here with if in "R is the number if individuals"

Line 204: Which figure this refers to isn't defined here.

Figure 1: Can the authors add a caption explanation for panels f-h? Also, is there a simple explanation for the non-linearities in Fig. 1d? Identifying the trajectories shown in panels a and b on the surface plots in panels c – h could provide additional intuition for the implications of different peak proportion infected and peak timing.

Figures 1-2: Matching color schemes for surface plots of wintertime peak proportion in infected could better allow for comparisons to be made between Fig. 1e and Fig. 2b.

Reviewer #3 (Remarks to the Author):

This submission is a nice paper that describes an approach to disentangle the likely involvement of climatic variables in SARS-CoV-2 incidence for a select number of locations. A challenge to this

important question is that there are limited evidence of the effects, and one can over-extrapolate from current observations. I think the authors have done a good job of analysing the question, but avoiding over-extrapolation. I provide a few suggestions in the analysis, hopefully to strengthen the approach and provide a bit more clarity.

Comments:

R₀ estimates:

1. So using epiestim and the functional equation seems ok.

2. It's currently unclear to me when 'NPI efficacy' was included in the model, ie for the whole time series or from a specified date? While I appreciate that 'the reality' is difficult to know, NPI are unlikely to be implemented for the whole time series, and not with equal measure, for example there is usually a (short) time from the first case until NPI is strongly implemented, and this may be regarded as an opportunity to estimate R₀ in the absence of NPI, and conversely NPI were relaxed pretty much everywhere at some point (eg summer). Consequently, if NPI are assumed to be constant but unknown, whereas in reality it is variable in the time series, it seems as though this will greatly affect the R₀ estimate. I think you should include at least further details for clarity of the approach and a discussion of potential issues and perhaps some sensitivity analysis.

3. The climate variable that you use is humidity, but maybe the main impact is temp or UV, or something else. How sensitive are your findings to fixing against humidity only? This might be a matter of exploring collinearity of humidity and the others to check.

Kath O'Reilly

Validation

1. I think there is value in comparing proportion recovered in the model variants in Figure 1 to available serology. I understand that there will be huge caveats with approach, but I) how much does recovered vary between scenario ii) does this help throw out some scenarios as being less plausible because they don't match the serology? It might be as simple as placing a dashed line on F1 c-h indicating the serology (assuming it exists)

I don't understand why F4 is not included as a result? Displaying the pie chart is actually quite useful. However, please provide details of how this is calculated, and what you mean by weather and climate specifically.

"Second, we do not directly estimate the climate sensitivity of SARS-CoV-2. Studies exploring this relationship using case data have yet to find a conclusive result". I think while there is a lot of analysis out there with differing results, I think a consensus is starting to emerge from experimental studies and some ecological studies that SARS-Cov-2 is moderately affected by cold and dry conditions <https://www.nature.com/articles/d41586-020-02972-4>

REVIEWER COMMENTS

Reviewer #1 (Remarks to the Author):

Convincing study of the potential effects of absolute humidity on Covid-19 transmission dynamics, allowing for the effects of NPI and population susceptibility.

We thank the reviewer for their helpful suggestions on improving the manuscript.

The meaning of Figure 2 could be made clearer by editing the caption as follows

The effect of *climate*,
changing susceptibility and NPIs on peak proportion infected...

Thank you for the suggestion, the caption has been updated.

and in line 123:

... meaning that secondary peaks in the climate scenario are lower than the constant R0 scenario for these locations
(add cross reference to Fig 2b)

We have added this Figure reference.

Further discussion of the implications of the model for Southern Hemisphere countries would be of interest.

We now include model results for Victoria, Australia in Fig. S4. Notably, we estimate that R0 for this location was much greater than 1 in July 2020. Our model projection suggests a substantial outbreak could have occurred, however, a second lockdown enforced in August 2020 successfully curbed the spread of the disease. This example illustrates our broader point on the importance of NPIs in limiting the spread. We now discuss this in the text.

“While our model assumes a constant reduction in R0 due to NPIs, in reality policy makers likely adapt to rising case numbers by enforcing stricter measures. For instance, we estimated an $R_0 > 1$ in Victoria, Australia in July and projected a much larger outbreak than has been observed to date (Supplementary Fig. 4). A second lockdown, enacted in August, substantially curbed this outbreak.”

Discussion

Consider commenting on the proposed physical explanations for effects of absolute humidity on viral transmission. In particular, airborne transmission of respiratory viruses is thought to be enhanced by desiccation of exhaled particles in dry air. This decreases particle aerodynamic diameter and thereby increases airborne residence time [3,4].

There are several hypothesis linking specific humidity to SARS-CoV-2 transmission. We now mention these explicitly in the text.

Specific humidity, in particular, has been shown to be important for influenza and RSV transmission \cite{shaman2009absolute, shaman2010absolute, baker2019epidemic}, with low specific humidity correlated with increased virus survival for influenza. Evidence suggests that humidity may also play a role in determining airborne droplet size and hence residence time in the air \cite{yang2011dynamics}.”

Minor comment

NPI= non pharmaceutical intervention, rather than control

Thank you for catching this! Now corrected.

Ref 10 is incomplete

This reference has been updated.

Simon Hales

Reviewer #2 (Remarks to the Author):

Summary: The authors use an epidemiological model for SARS-CoV-2 with an assumed climate-dependent transmission rate to project the outbreak into the future in New York City and other locations under varying levels of non-pharmaceutical interventions and population susceptibility. This is a clearly written paper on a topic of great interest and importance, but I have some concerns with it in its current form.

We thank the reviewer for their very helpful suggestions and comments.

Major comments: My main concern with the study as it is currently written is with regards to some of the underlying assumptions regarding climate sensitivity of SARS-CoV-2, and the lack of justification and exploration of the importance of these assumptions.

1) Assuming climate sensitivity of SARS-CoV-2 is the same as HKU1

The first and most important assumption from my perspective is the assumption that the climate sensitivity of SARS-CoV-2 will match that as estimated for another betacoronavirus, HKU1. The introduction (lines 52-53) notes to “see Discussion for the effect of varying climate sensitivity”, but I could only find a sentence or two noting that there has yet to be a conclusive result from studies exploring the climate sensitivity of SARS-CoV-2 and so the authors have assumed it follows that of HKU1. In the authors’ previous study that this work builds on (Baker et al 2020, Science), they estimate that HKU1 shows the strongest sensitivity to climate of the three viruses compared, while the betacoronavirus OC43 shows the weakest climate sensitivity. If these two viruses represent the two extremes (strong climate sensitivity versus weak climate sensitivity), are there reasons that we should expect SARS-CoV-2 to match the pattern of strong climate sensitivity? If, instead, it’s equally likely to follow the weak climate sensitivity of OC43 (or to fall somewhere in the middle), I think these scenarios should be given more exploration here. I appreciate that the authors have done the uncertainty decomposition and have included varying climate sensitivity as a factor in Figure 4, but I don’t think this necessarily makes clear the importance of this assumption or shows the effect that assuming a different climate sensitivity would have on the results. In other words, I’m wondering how different Figure 1 and the results section would look if the authors had assumed SARS-CoV-2 follows the climate sensitivity of OC43 instead of HKU1?

We agree with the reviewer that we should also consider the OC43 sensitivity. In Fig. S6 (reproduced below) we show the results from Fig. 1e,h using the climate sensitivity of OC43 (plotted using the same color gradient and with HKU1 results side-by-side for comparison). The relative effect of climate on wintertime peak size is much smaller for

the OC43 scenario. For this set of parameters, the additional effect of climate is now maximum at 0.02, under half the size of the HKU1 maximum of 0.05.

There is an additional complexity worth considering. In the OC43 scenario, R_0 remain high throughout the year (with less of decline in the summer months, as expected in New York with the HKU1 scenario). This means the measured relative efficacy of NPIs in July is greater in the OC43 scenario: NPIs are now reducing R_0 by 55% below what it would have been in the summer months to reach $R_0 = 1$, as opposed to the 35% reduction in the HKU1 scenario. In new Fig. S7, we show the results over this larger range of NPI reduction. In this case, we change the color scale to span the max/min of relative peak size observed for OC43. Over this range, we reveal the same pattern as observed in the HKU1 results. However, the maximum relative peak size remains smaller at 0.02.

We have added a section in the discussion to explain these results.

We have also added a line earlier in the paper when we introduce the HKU1 scenario “This betacoronavirus was found to be the more sensitive to climate in our recent work and so our simulations reveal the upper bound on a possible climate effect.”

2) R0max and R0min

I was also hoping for a bit more justification for the assumed R0max and R0min values used in Eq. 1 beyond the explanation that they were previously estimated in the Baker et al 2020 and Kissler et al 2020, as the choices for these values can greatly affect how strong the realized climate effect is on the virus. From my reading of Baker et al 2020, it seemed that R0max of 2.5 was used as a conservative lower estimate early in the epidemic; does this still hold as the likely maximum R0 value for SARS-CoV-2? Similarly, it seems that R0min of 1.5 was used because other viruses have shown up to a 40% decrease due to environmental factors. Do we have any reasons to suspect that SARS-CoV-2 follows this same 40% pattern? Overall, even though it may be somewhat repetitive of what was stated in the Baker et al 2020 Science paper, I think this section could benefit from some added methods descriptions that state the assumptions underlying this equation.

We agree with the reviewer. We have now updated the methods description to better explain the assumptions behind the R0 choices. There have been substantial effort to estimate R0 and several review studies also published of this work. Majumder and Mandl review early studies finding that peer-reviewed papers had an average R0 of 2.54 and non-peer reviewed preprints had an average R0 of 3.02. Liu et al 2020 also reviewed the literature, finding a minimum estimate of R0 of 1.4 and maximum estimate of 6.49, with a mean of 3.28 and median 2.79. Hilton and Keeling use an R0 of 2.4 and review several estimates from the literature, finding a minimum of 2 and maximum of 3.11. In summary these studies broadly point to an R0 in the range of 2-3. As many of these studies use early (January/February) case data from China, when specific humidity is at a minimum, we assume this range of R0s reflect R0max. R0min is harder to uncover based on current research, though using 1.5 does match with the lower bound of estimates reviewed in Liu et al.

We have updated the text to explain our reasoning in choosing these values of max and minimum R0. We also show the results using a higher R0max of 3 in Fig. S7. This does not change our results.

Updated methods text : “Most estimates of R_0 for SARS-CoV-2 lie in the range of 2-3 (Liu2020reproductive, Majumder2020early, Hilton2020estimation), though some studies have estimated an R_0 as high as 6.49 and as low as 1.44 (Liu2020reproductive). We set $R_{0max} = 2.5$ as a conservative upper bound. In prior work we set $R_{0min} = 1.5$ to reflect the 40% reduction in transmission observed due to climate in studies of other viruses (Kissler2020projecting, Baker2020susceptible). We find this value matches lower bound estimates of R_0 (Liu2020reproductive), however, the strength of the climate effect for SARS-CoV-2 remains uncertain. Results using a higher R_{0max} are shown in Supplementary Figure 8. “

3) Application of the model to global locations and potential alternative drivers

It is not clear to me if the same model used to predict wintertime outbreaks in New York City should be applied to a range of global cities, including some in the tropics. From what I understand, the climate sensitivity relationship (Eq. 1) that is applied here to global locations (Fig. 2) was fit in Baker et al 2020 Science to surveillance data from the United States and is driven by changes in specific humidity. While specific humidity may be the main climate driver of betacoronavirus dynamics in the United States or temperate regions, is it a fair assumption that it is also the main climate driver in some of the tropical cities modeled here in Figure 2? For example, in Tamerius et al 2013 (PLoS Pathogens), they found that while influenza cases peaked under cold and dry conditions in temperate areas, tropical areas showed a different, u-shaped response where influenza cases peaked at very low levels of specific humidity but also at very high levels of specific humidity. It

therefore seems very possible that other climate drivers that have been suggested to be important in the tropics for transmission of other respiratory viruses like influenza, such as precipitation or ultraviolet radiation, could be more important factors than specific humidity for SARS-CoV-2 climate dependence in some of the cities modeled here. The authors state that “tropical locations experience minimal difference in the climate versus constant R0 scenario given the relatively mild seasonal variations in specific humidity in the tropics”, but can they please provide some justification for why they expect specific humidity to be the climate driver in these cities, and why they’d expect to see the same transmission - specific humidity relationship that was estimated in the United States hold across the globe?

Several respiratory viral infections exhibit a pattern of strongly seasonal dynamics at high latitudes and more persistent outbreaks closer to the tropics. This has been observed for influenza [Viboud 2006] as well as RSV [Baker 2019]. While there is generally limited surveillance of the endemic betacoronaviruses, particularly in tropical locations, a study in Malaysia also finds persistent cases [Al-Khannaq 2016] suggesting this tropical/temperate divide persists for these viruses.

In prior work on RSV [Baker 2019], we found that a common set of environmental drivers can explain the gradient in RSV dynamics observed from temperate to tropical locations. We found that specific humidity dominates dynamics but in regions with low variability in specific humidity i.e. tropical locations, a more modest effect of precipitation can cause a peak in cases close to the rainy season. However, locations with the rainfall peak still experienced relatively persistent case numbers throughout the year, suggesting the boost to transmission from rainfall was not of the same magnitude as the specific humidity effect.

As the reviewer notes, it is possible that a secondary variable, such as precipitation, could be important in tropical locations. However, the persistent dynamics observed in Malaysia for HKU1/OC43 suggest that the effect of this secondary variable may be moderate. Nonetheless, we agree it is a limitation of the work that we only use US data to characterize our model.

We have now discussed this limitation to our model in the manuscript. Noting that our model can capture the persistence in Malaysia in Fig. S9.

“Third, evidence from other respiratory viral pathogens implies that tropical locations may experience distinct climate drivers and dynamics of infection \cite{baker2019epidemic, viboud2006influenza, tamerius2013environmental}. Given our model is parameterized by fitting to US data, we may not have captured the full suite of possible climate drivers in these locations. While data from tropical climates is limited, evidence from Malaysia suggests persistent year-round infections for the endemic betacoronaviruses \cite{al2016molecular}, a feature our model is able to capture (Supplementary Fig. 9). However, possible secondary climate drivers will not be accounted for the model and could bias our results for tropical locations.”

Relatedly, given that other climate drivers may be relevant and have similar average seasonal patterns but different interannual variation, is there evidence that the interannual variation in specific humidity drives interannual variation in other coronaviruses? The authors argue that there could be large effects of interannual variation in specific humidity in NYC on interannual changes in cases, but the climate-sensitivity is estimated using the average seasonal pattern of specific humidity. Alternatively, if there’s no signal of an effect of interannual variation of specific humidity on interannual variation in HKU1 or OC43 cases over the last years, should we expect an effect for SARS-CoV-2?

The betacoronavirus case data used to parameterize the SIRS model is at a very coarse spatial resolution (the census region level i.e. groupings of multiple states) and a relatively short time series (5 years). The lack of spatial and temporally resolved data limits our ability to identify the effect of interannual variation. This is now noted in the discussion and in more detail in the methods.

Minor comments

Line 20: if the abbreviation is NPIs, I think this should read “interventions” instead of “controls”.

Thank you for catching this mistake! Now corrected.

Line 22-23: “Our results imply a role for meteorological forecasts in projecting outbreak severity”. Does this statement rely on the assumption that SARS-CoV-2 climate dependence will be as strong as HKU1? If SARS-CoV-2 climate dependence instead is similar to OC43, would meteorological forecasts still be useful?

Our final Figure, Figure 4, does not rely solely on the HKU1 scenario. In this case, climate sensitivity is varied between OC43 and HKU1 values. The portion assigned to weather variability in this figure reflects the contribution to wintertime peak variance across this range of scenarios.

However, we agree with the reviewer that lines 22-23 overly emphasize a role for meteorological forecasts, given that climate sensitivity may be uncertain. Our main finding from Figure 4 is that the strength of NPIs are likely the dominant driver of wintertime peak size. We have updated the abstract to better reflect this meaning.

“Our results suggest that the strength of NPIs remain the greatest determinant of future outbreak size. While we find a small role for meteorological forecasts in projecting outbreak severity, reducing uncertainty in epidemiological parameters will likely have a more substantial impact on generating accurate predictions.”

Lines 37-38: “Endemic coronaviruses have also demonstrated sensitivity to the climate in both laboratory studies [8]

and at the population level [7].” Reference 7 relates to RSV – are there references for coronavirus that could be used here?

Sorry, this was a mistake. We meant to cite [1] (Baker 2020). Another study investigating the seasonality of the endemic coronaviruses is Neher 2020 – while these authors do not explicitly relate the seasonality back to possible climate drivers, they uncover distinct seasonal patterns in transmission and speculate “this scenario, with maximal β in mid-winter, is also more compatible with climate variation around the year.” We now also cite this study.

We note that there have been relatively few studies of the climate drivers of the endemic coronaviruses using population scale data, especially when compared to other respiratory pathogens such as influenza. This is likely due to the lack of detailed data streams on these infections: because the endemic coronaviruses cause relatively mild symptoms, surveillance of these pathogens has not historically been a priority.

Lines 44-45, 53-54: The emphasis on effects in “northern hemisphere” locations seems misplaced given the broader focus on global cities later in the paper.

We agree with the reviewer. We have removed the first instance referring to the northern hemisphere. When we refer to the northern hemisphere at the end of this paragraph, we are partly explaining our focus on New York for the first set of analyses so we have retained this instance. We have also expanded discussion of the Southern Hemisphere (including cases in Victoria) in line with suggestions from the first reviewer.

Lines 61-64: Further explanation of the “climate-driven model” would help here to explain the assumptions inherent in the subsequent results.

We have added a sentence to clarify two factors about the model: that it is dependent on specific humidity and based on betacoronavirus HKU1. As noted in response to Main point 1), we have also highlighted the implications of using HKU1 in the preceding paragraph.

“The model assumes the climate sensitivity of betacoronavirus HKU1 and that seasonal variations in transmission are driven by specific humidity.”

Lines 109-110: Is this result that secondary peak timing for New York is clustered in the winter months in the climate scenario (but can occur at a wide range of times in the constant scenario) almost a foregone conclusion because of the way climate-dependence is incorporated into the authors’ model?

When susceptibility is high, and NPIs are limited, large outbreaks occur in both the climate and the constant scenario in the late summer, early autumn. The effect of climate in this case is not strong enough to constrain the spread to winter months.

Lines 127-128: “This result is more striking in regions with a large seasonality in specific humidity” – can the authors add a note to the end of this sentence (and/or in the Fig. 2 caption) which cities this refers to specifically.

In general, we are referring to the northern hemisphere and southern hemisphere locations, where seasonality in specific humidity is high. We have now updated Figure 2, to make clear which are the northern hemisphere, tropical and southern hemisphere locations. We have also updated the text to refer to 3 highly seasonal locations: Delhi, New York and Johannesburg.

Lines 130-140, 168-169: Is there any analogous evidence from the other betacoronaviruses to help support the argument that interannual variation could have large effects on interannual changes in cases? See major comment 3 for more on this.

As above, we note that the US betacoronavirus data is at a coarse spatial resolution with a short time horizon, limiting our ability to identify the signal of interannual specific humidity variability on cases. This is now noted in the methods.

Line 193: typo in “sensitivity”.

Thank you – this has been corrected.

Lines 198-200: L in Eq. 2 is not defined in text, and R, which is defined, does not appear in either equation. I assume it's just that $N = S + I + R$, but that would be worth stating.

We have clarified this.

Line 199: I think there's a typo here with if in "R is the number if individuals"

Corrected!

Line 204: Which figure this refers to isn't defined here.

Corrected!

Figure 1: Can the authors add a caption explanation for panels f-h? Also, is there a simple explanation for the non-linearities in Fig. 1d? Identifying the trajectories shown in panels a and b on the surface plots in panels c – h could provide additional intuition for the implications of different peak proportion infected and peak timing.

We have now updated the caption of Figure 1 to include an explanation for f-h (we apologize, this was mistakenly left off in the previous draft). The non-linearities in Figure 1d occur with changing susceptibility. High susceptibility means more cases, however, lower susceptibility means the outbreak is more driven by climate and therefore more concentrated in the winter. The trade-off between these two factors leads to the nonlinearity in Fig 1d.

We have now added trajectory locations to Figures 1c-h.

Figures 1-2: Matching color schemes for surface plots of wintertime peak proportion in infected could better allow for comparisons to be made between Fig. 1e and Fig. 2b.

In Fig. 2b we also include southern hemisphere locations where climate factors help lower R_0 below current values. Fig. 2b therefore has a much larger range than Figure 2b. It was not possible to put these on the same scale while retaining key interpretability of Fig. 1e.

Reviewer #3 (Remarks to the Author):

This submission is a nice paper that describes an approach to disentangle the likely involvement of climatic variables in SARS-CoV-2 incidence for a select number of locations. A challenge to this important question is that there are limited evidence of the effects, and one can over-extrapolate from current observations. I think the authors have done of good job of analysing the question, but avoiding over-extrapolation. I provide a few suggestions in the analysis, hopefully to strengthen the approach and provide a bit more clarity.

Many thanks for the helpful comments on the paper!

Comments:

R_0 estimates:

1. So using epiestim and the functional equation seems ok.

2. It's currently unclear to me when 'NPI efficacy' was included in the model, ie for the whole time series or from a specified date? While I appreciate that 'the reality' is difficult to know, NPI are unlikely to be implemented for the whole time series, and not with equal measure, for example there is usually a (short) time from the first case until NPI is strongly implemented, and this may be regarded as an opportunity to estimate R_0 in the absence of NPI, and conversely NPI were relaxed pretty much everywhere at some point (eg summer). Consequently, if NPI are assumed to be constant but unknown, whereas in reality it is variable in the time series, it seems as though this will greatly affect the R_0 estimate. I think you should include at least further details for clarity of the approach and a discussion of potential issues and perhaps some sensitivity analysis.

We apologize for the unclear explanation of the NPIs. When we run the model for New York (e.g. in Figure 1), we input the unaltered R_0 estimated from epiestim for the first half of the time series, up until July 2020 (when we began

this analysis – i.e. the present, at that time). The estimated R0s include a period with high values, presumably reflecting the period before NPIs came into force, and then the subsequent decline due to NPI measures.

For the second half of the time series, we extrapolate by setting R0 equal to the average estimated July values. For New York, we found the first two weeks of July had an average R0 (based on epiestim) of 1.04. To turn this value into a measure of NPI efficacy, we calculate the percentage reduction based on our predicted R0s for this time of year (based on our climate-SIRS model). Our model suggests an average R0 of 1.61 for the first two weeks of July – so we get an efficacy of $1 - 1.04/1.61 = 35\%$. We need to calculate this efficacy because in the climate scenario we assume our R0 oscillates with climate – but at this percentage reduction determined by NPIs.

Finally, we allow this percentage reduction (or NPI efficacy) to vary between 0 – 35%, across scenarios, assuming we don't know how effective NPIs might be. We have now clarified this in the main text and also updated the methods.

There are of course many possible additional scenarios for NPIs in the future. However, in order to retain clarity in the effect of climate, we limit our exploration of scenarios to only varying the percentage reduction due to NPIs. We have now noted in the discussion the limitation of this approach, and the reality that many policy makers adapt to rising cases by increasing NPI measures. This means our scenarios are not completely realistic, however, further highlights our point that NPI measures can and will be leveraged to mitigate future outbreaks and that the future of the pandemic is under human

3. The climate variable that you use if humidity, but maybe the main impact is temp or UV, or something else. How sensitive are you findings to fixing against humidity only? This might be a matter of exploring collinearity of humidity and the others to check.

There is a very high correlation (98%) between average weekly temperature and specific humidity (now shown in Fig. S10 for New York city). These variables are functionally dependent via the Clausius Clapeyron relation. It is therefore hard to disentangle the effect of the two variables with the limited dataset available and likely fitting our model to temperature data would uncover a similar result. UV and specific humidity are also correlated (75%). This has now been noted in the discussion and methods.

“We fit our model to the mean seasonality in specific humidity which follows a sinusoidal pattern tightly correlated with temperature (Supplementary Fig. 10). Fitting to temperature would likely give similar results (though with scaled values of β). Other climate variables seasonally correlated with specific humidity, such as patterns of ultraviolet radiation, may have a similar effect. However, we retain specific humidity as a driving variable due to prior understanding of its role in respiratory pathogen transmission [shaman2009absolute, shaman2010absolute, baker2019epidemic].”

Kath O'Reilly

Validation

1. I think there is value in comparing proportion recovered in the model variants in Figure 1 to available serology. I understand that there will be huge caveats with approach, but I) how much does recovered vary between scenario ii) does this help throw out some scenarios as being less plausible because they don't match the serology? It might be as simple as placing a dashed line on F1 c-h indicating the serology (assuming it exists)

We thank the reviewer for this suggestion. In recently published work, Stadlbauer (2020) estimate a seroprevalence of 20% in New York. We have added a line on Fig.1 c-h to show this.

I don't understand why F4 is not included as a result? Displaying the pie chart is actually quite useful. However, please provide details of how this is calculated, and what you mean by weather and climate specifically.

We agree with the reviewer. We have now included F4 in the results section of the paper and improved description of this figure in the main text.

"Second, we do not directly estimate the climate sensitivity of SARS-CoV-2. Studies exploring this relationship using case data have yet to find a conclusive result". I think while there is a lot of analysis out there with differing results, I think a consensus is starting to emerge from experimental studies and some ecological studies that SARS-Cov-2 is moderately affected by cold and dry conditions <https://www.nature.com/articles/d41586-020-02972-4>

We have updated this text to reflect the growing consensus.

"Second, we do not directly estimate the climate sensitivity of SARS-CoV-2. Studies exploring this relationship using case data have yet to find a definitive result, however, there is increasing evidence that the virus is moderately affected by cold, dry conditions \cite{Morris2020, Smit2020}."

References

Al-Khannaq, Maryam Nabeel, et al. "Molecular epidemiology and evolutionary histories of human coronavirus OC43 and HKU1 among patients with upper respiratory tract infections in Kuala Lumpur, Malaysia." *Virology journal* 13.1 (2016): 33.

Hilton, Joe, and Matt J. Keeling. "Estimation of country-level basic reproductive ratios for novel Coronavirus (SARS-CoV-2/COVID-19) using synthetic contact matrices." *PLoS computational biology* 16.7 (2020): e1008031.

Liu, Ying, et al. "The reproductive number of COVID-19 is higher compared to SARS coronavirus." *Journal of travel medicine*

Stadlbauer et al. "Repeated cross-sectional sero-monitoring of SARS-CoV-2 in New York City." *Nature* (2020)

Majumder, Maimuna S., and Kenneth D. Mandl. "Early in the epidemic: impact of preprints on global discourse about COVID-19 transmissibility." *The Lancet Global Health* 8.5 (2020): e627-e630.

Viboud, Cécile, Wladimir J. Alonso, and Lone Simonsen. "Influenza in tropical regions." *PLoS Med* 3.4 (2006): e89.

REVIEWERS' COMMENTS

Reviewer #2 (Remarks to the Author):

The authors have done a commendable job addressing reviewer comments and suggestions.

I am glad to see the new analysis using OC43 instead of HKU1, as it does provide quite different results, and the new section in the discussion helps speak to this. Also, the new line earlier in the paper that states the HKU1 scenario reveals an upper bound of a possible climate effects is very useful.

The authors have also addressed all of my other major and minor comments.

Reviewer #3 (Remarks to the Author):

Thank you for fully responding to my comments. And a good rebuttal to reviewer 2.

REVIEWERS' COMMENTS

Reviewer #2 (Remarks to the Author):

The authors have done a commendable job addressing reviewer comments and suggestions.

I am glad to see the new analysis using OC43 instead of HKU1, as it does provide quite different results, and the new section in the discussion helps speak to this. Also, the new line earlier in the paper that states the HKU1 scenario reveals an upper bound of a possible climate effects is very useful.

The authors have also addressed all of my other major and minor comments.

We thank the reviewer for their helpful comments which have greatly improved the manuscript.

Reviewer #3 (Remarks to the Author):

Thank you for fully responding to my comments. And a good rebuttal to reviewer 2.

Many thanks for the helpful comments!